# Semi-Supervised Defect Detection Method with Data-Expanding Strategy for PCB Quality Inspection

**DOI:** 10.3390/s22207971

**Published:** 2022-10-19

**Authors:** Yusen Wan, Liang Gao, Xinyu Li, Yiping Gao

**Affiliations:** School of Mechanical Science and Engineering, Huazhong University of Science and Technology, Wuhan 430074, China

**Keywords:** PCB defect detection, semi-supervised learning, deep learning, data expanding

## Abstract

Printed circuit board (PCB) defect detection plays a crucial role in PCB production, and the popular methods are based on deep learning and require large-scale datasets with high-level ground-truth labels, in which it is time-consuming and costly to label these datasets. Semi-supervised learning (SSL) methods, which reduce the need for labeled samples by leveraging unlabeled samples, can address this problem well. However, for PCB defects, the detection accuracy on small numbers of labeled samples still needs to be improved because the number of labeled samples is small, and the training process will be disturbed by the unlabeled samples. To overcome this problem, this paper proposed a semi-supervised defect detection method with a data-expanding strategy (DE-SSD). The proposed DE-SSD uses both the labeled and unlabeled samples, which can reduce the cost of data labeling, and a batch-adding strategy (BA-SSL) is introduced to leverage the unlabeled data with less disturbance. Moreover, a data-expanding (DE) strategy is proposed to use the labeled samples from other datasets to expand the target dataset, which can also prevent the disturbance by the unlabeled samples. Based on the improvements, the proposed DE-SSD can achieve competitive results for PCB defects with fewer labeled samples. The experimental results on DeepPCB indicate that the proposed DE-SSD achieves state-of-the-art performance, which is improved by 4.7 mAP at least compared with the previous methods.

## 1. Introduction

With the wide use of electronic products, the manufacture of PCBs has shown increasing importance in modern manufacturing [1,2,3]. However, the defects in PCBs may lead to circuitry burnout. Thus, PCB defect detection, whose core purpose is to locate and classify the defects in PCB samples, is important, and developing an efficient PCB defect detection method can improve the quality and efficiency of PCB manufacturing and is helpful to modern industrial manufacturing. Because of the requirements for efficiency, a useful method needs to be capable to accurately and promptly detect defects.

Traditionally, PCB defect detection depended on the manual inspection of experts. With the development of intelligent sensors, PCB image sample collection became easier. Therefore, defect detection methods based on machine vision became mainstream. Wang et al. combined image registration, image segmentation, drill numbering, drill contrast, and defect displays to inspect the defects in PCBs, which achieved great improvement [4]. Moreover, AJ et al. used a genetic algorithm to search for the template positions and used the normalized cross-correlation (NCC) template-matching approach to locate the defect positions [5]. Similarly, Gaidhane et al. proposed a feature-free method to measure the similarity between a scene image and corresponding template image [6]. Zhong et al. realized rapid defect detection by three successive procedures, namely image contrast enhancement, obtainment of the standard templates to weaken the texture interferences, and probability calculation of the background feature of each pixel in the edge region between the background and defect [7]. With the development of machine learning (ML), some researchers attempted to leverage ML algorithms to detect defects. Yuk et al. generated a weighted kernel density estimation (WKDE) map weighted by the probabilities to locate the defects [8]. Furthermore, some researchers proposed an online PCB defect detection method by leveraging the knowledge from machine vision [9].

With the rapid development of artificial intelligence, especially deep learning (DL) and computer vision, researchers attempt to use object detection methods based on DL to detect defects in PCBs to achieve high detection accuracy and automatic defect detection [10,11,12]. Some researchers developed a new deep neural network (DNN) architecture, especially for PCB defect detection. Luo et al. proposed a decoupled two-stage object detection framework for PCB defect detection [13]. He et al. proposed a real-time PCB defect detector that could make use of both the detect-free template images and images to be checked [14]. Gao et al. used a feature collection and compression network to merge multiscale feature information and used a new Gaussian weighted pooling to replace the ROI pooling [15]. These methods achieve automatic defect detection but still need the researchers to manually develop a DNN. Thus, other researchers are trying to apply the classical object detection methods to PCB defect detection and improve them. Ding et al. proposed a TDD-net for the tiny defect in PCBs, which used a faster R-CNN as a baseline model, and proposed a strategy to mine the hard samples online [16]. Jin et al. applied deformable DETR on PCB defect inspection [17]. Though they achieved accurate defect detection, these methods’ detection speed could not satisfy the requirements of industrial manufacture. In actual manufacturing, single-stage detectors are widely used because they can rapidly detect defects. Jin et al. attempted to use EfficientDet [18] to detect defects and use K-means clusters to predict more accurate anchors [19]. Another kind of single-stage object detection method is You-Only-Look-Once (YOLO) [20,21], which is the most popular object detection method used in industrial manufacture especially PCB defect detection. Xin et al. improved a YOLOv4 and achieved better performance on the PCB dataset [22]. In addition, Adibhatla et al. used the newest YOLO model, YOLOv5, to detect the defects in PCBs [23]. There are also some researchers using different kinds of YOLO models to detect PCB defects [24,25,26,27]. The defect detection methods based on DL have achieved automatic deep feature extraction and defect location with high accuracy.

However, the detection models based on DL require a huge number of samples with high-level labels. It is difficult to collect plenty of labeled samples in real-world applications mainly because of two reasons: (1) Labeling the PCB samples is a time-consuming and exhausting task. Thus, there are often a few labeled samples with lots of unlabeled samples [28]. (2) It is impracticable to integrate several small-scale datasets to generate a large-scale PCB dataset because these datasets are collected by different kinds of sensors and the backgrounds and defects are different, as shown in Figure 1. Therefore, only a small number of labeled samples in the given dataset can be used by most existing PCB defect detection methods, which causes the detection accuracy to be too low and other samples to be wasted.

To address this problem, some researchers proposed semi-supervised defect detection methods such as SSD [14]. These methods can leverage both labeled and unlabeled samples to train the detection model by generating pseudo labels for unlabeled samples and improve the detection accuracy a lot when the number of labeled samples is small. However, the detection accuracy of SSD trained on small amounts of labeled samples is 9.7% lower than that on huge numbers of labeled samples. This is because the training process will be disturbed by the unlabeled samples when the proportion of labeled samples is too small.

Therefore, except for enabling the model to leverage unlabeled samples, this paper tried to find a semi-supervised defect detection method that is robust against the disturbance by unlabeled samples. As shown in Figure 2, the paper proposed a **s**emi-**s**upervised **d**efect detection method with a **d**ata-**e**xpanding strategy (DE-SSD), containing a detection model, an SSL framework with a **b**atch-**a**dding strategy (BA-SSL), and a data-expanding (DE) strategy. This method can train a detection model by using both labeled and unlabeled data and expand the given dataset by using the samples from other datasets. In this work, the researchers selected YOLOv5 as the detection model to detect the defects in PCB samples because of its high detection accuracy and high FPS. The BA-SSL framework is proposed to enable the method to leverage the unlabeled samples to train the detection model by pseudo labels without too much disturbance. It contains a batch-adding strategy for unlabeled samples and a dynamic threshold strategy for the pseudo-label generation to reduce the disturbance. The DE strategy is proposed to leverage the labeled samples from other datasets to expand the training data by eliminating the difference between samples collected by different sensors, which can limit the proportion of unlabeled samples and reduce the disturbance. The whole training process is shown in Figure 2 and contains three steps. The first is training the model on labeled samples, the second is adding the unlabeled samples batch by batch, and the third is training the model on both labeled and unlabeled samples.

The main contributions are as follows:(1)This paper proposed a BA-SSL framework for PCB defect detection to leverage both the labeled and unlabeled data, which contains a proposed dynamic threshold strategy and a proposed batch-adding strategy for unlabeled data.(2)This paper proposed a data-expanding strategy to leverage the samples from auxiliary datasets as negative samples to expand the target dataset by reducing the difference in backgrounds and sizes of samples from different datasets.(3)This paper proposed a **s**emi-**s**upervised **d**efect detection method with a **d**ata-**e**xpanding strategy (DE-SSD) for PCB defect detection by integrating the aforementioned two points. The researchers test it on the DeepPCB dataset and use PKU-PCB as an auxiliary dataset. It achieves mAP 98.4 with only 50 labeled samples and is a state-of-the-art method.

The remainder of this paper is organized as follows. Section 2 introduces the datasets used in this paper and the problems this paper tends to address. Section 3 presents the details of the proposed method. Section 4 gives the experimental results and analyses. Section 5 presents the results of the ablation study. Finally, Section 6 concludes this article and looks into future work.

## 2. Datasets and Problem Description

This section introduces the datasets used in this work, including a target dataset and an auxiliary dataset, compares them to analyze the difference, and gives the problem this work aims at.

### 2.1. Target Dataset

The target dataset is the main dataset that detection models need to detect. As this paper proposed a DE-SSD method to leverage unlabeled samples to improve detection accuracy, the researchers need to conduct experiments on a public benchmark to compare the proposed method with other methods. In this work, the researchers selected a widely used dataset, DeepPCB, as the target dataset [14] because there are lots of previous works conducting experiments on it, the types of defects in it are representative, and the samples in it were collected by a linear-scan CCD, which is widely used in practical manufacturing. Therefore, the researchers can fairly and fully compare the proposed method with other methods.

In the DeepPCB dataset, there are a total of 1500 samples, of which 1000 samples are used to train the model and the others are used to test the model. Every defect image has a corresponding template image. There are six common types of PCB defects in samples: open, short, mouse bite, spur, spurious copper, and pinhole, which are the most representative types of PCB defects and are widely used in most of the previous works [16]. A few of them are real defects, and the others are artificial defects added by the researchers, which account for around 3 to 12 defects in each sample.

The PCB samples in DeepPCB are collected by a linear-scan CCD. When the CCD scanner scans the PCBs, the linear CCD divides the scanned PCBs into lines, and the sensors receive the reflected light from the lines and generate electrical signals according to the light intensity. The linear-scan CCD scans the PCBs and lines and combines the lines into RGB images. Then, the defect-free template images and defect images are manually classified and cut into sub-images with a size of 640×640. The images are transmitted into binary with a carefully selected threshold.

The defects in samples are shown in Figure 3. Open defects occur when the trace breaks or when the solder is only on the pad. Short defects occur when the distance between traces is too small or the leads of components are not trimmed. The mouse bite defects are the sharp edges left over after depanelization. The spur defects and spurious copper defects may cause the interruption of normal flow. The pinhole defects are caused by PCBs outgassing during soldering.

### 2.2. Auxiliary Dataset

Auxiliary datasets are used to improve the detection accuracy tested on the target dataset. In this work, the researchers selected the PKU-PCB dataset as the auxiliary dataset because its main structure is the same as that of DeepPCB. The PKU-PCB dataset is a synthetic PCB defect detection dataset released by Peking University, which contains 1386 PCB samples. The researchers collected the template images and manually added the defects. The width of each sample is longer than 3000 pixels. Each sample is an RGB image, which contains a whole PCB and several manual defects in it. There are six types of PCB defects in the samples: open, short, mouse bite, spur, spurious copper, and missing hole.

### 2.3. Difference and Analysis

The target and auxiliary datasets have two types of differences.

The main difference is the defect, including the types and numbers. There are five types in common: open, short, mouse bite, spur, and spurious copper, but the sixth type in PKU-PCB is “missing hole”, whereas the sixth type in DeepPCB is “pinhole”. In addition, in every sample of PKU-PCB, there are only about two to three defects, whereas the figure for DeepPCB is 3–12. The other difference is the background because of the collection methods. The samples of PKU-PCB are color images and have a green background, whereas those in DeepPCB are black-and-white images and have a white background.

### 2.4. Problem Description

As a result, although there are plenty of PCB samples, including a few labeled samples, lots of unlabeled samples in the target dataset, and plenty of samples from other datasets, the existing methods can only leverage a small part of them. If researchers try to leverage the unlabeled samples [14], the unlabeled samples will disturb the training process, and the detection accuracies on small numbers of labeled samples are significantly lower than those on large numbers of labeled samples. Moreover, the aforementioned difference prevents the researchers from using the samples from auxiliary datasets to improve the number of labeled samples.

Therefore, this paper tended to eliminate the accuracy gap between small and large numbers of labeled samples by addressing the aforementioned problems.

## 3. Proposed DE-SSD Method

To leverage the unlabeled data and auxiliary datasets, this paper developed a DE-SSD method, which contains a single-stage object detection model, a BA-SSL framework, and a data-expanding strategy.

This work selected YOLOv5, a classical object detection model, as the detection model because of its high detection accuracy and high FPS. The BA-SSL framework is based on pseudo labels, which contains an end-to-end training strategy, dynamic threshold strategy for pseudo-label generation, and batch-adding strategy for unlabeled data. As shown in Figure 2, BA-SSL contains three steps. Firstly, the detection model is trained on only labeled data. Then, the unlabeled data are put into the training process batch by batch. Finally, the detection model keeps being trained on a stable amount of data. In addition, the data-expanding strategy is proposed to expand the target dataset by generating negative samples according to auxiliary datasets. As shown in Figure 2, the samples from auxiliary datasets are cut into pitches and processed by a data augmentation strategy. In this work, the samples from auxiliary datasets were used as negative samples.

In this section, the structure of the object detection model, YOLOv5, is introduced first. Then, the training process of the proposed BA-SSL framework is given. Finally, detailed information about the proposed DE strategy is provided.

### 3.1. Structure of YOLOv5

In this part, this paper introduced the structure of YOLOv5, which was used as a detection model in this work. As shown in Figure 4, YOLOv5 can be divided into four parts: input, backbone, neck, and prediction [23].

At the input end, YOLOv5 has a mosaic data augmentation strategy and adaptive anchor size calculation based on K-means clusters. This strategy can make the detection model more robust by adding small targets, which is especially suitable for a dataset that contains plenty of small targets. Moreover, K-means clusters can automatically calculate and update the best size of anchors while training.

In the backbone, YOLOv5 contains several blocks, namely focus, CBL, CSP, and SPP. The focus block slips an original image to feature maps, combines them, and uses convolution kernels to generate a feature map. After the focus block, there are groups of alternating CBL and CSP1_X blocks. These blocks extract features from the original image. At the end of the backbone, there is an SPP block to improve the model’s receptive field.

The neck part of YOLOv5 has an FPN + PAN structure. This structure leverages the lesson from PANet and contains an FPN and a PAN. The FPN uses downsampling and delivers the strong semantic features top-down, whereas the PAN uses upsampling and delivers the strong location information bottom-up. This structure can improve the model’s ability for feature extraction.

At the output end, the model will output three feature maps in different sizes. While training, YOLOv5 uses cross-entropy (CE) loss as the loss function of the classification scores and complete intersection over union (CIOU) loss as the loss function of the bounding box.

YOLOv5 is the most advanced single-stage object detection method. In previous works, some researchers used it to detect defects in PCBs and it showed outperformance. Therefore, in this work, the researchers selected YOLOv5 as the detection model.

### 3.2. BA-SSL: SSL Framework with Batch-Adding Strategy

This paper proposed a **s**emi-**s**upervised **l**earning framework with a **b**atch-**a**dding strategy (BA-SSL) to train the detection model on both labeled and unlabeled data. It has an end-to-end training strategy, proposed dynamic threshold strategy, and proposed batch-adding strategy for unlabeled data. The end-to-end training strategy can train the detection model end-to-end, which contains three steps. The training process is shown as follows:(1)In the first step, the detection model is trained on only labeled data from the target and auxiliary datasets.(2)In the second step, both the labeled and unlabeled data are used to train the detection model, and the amount of unlabeled data increases with training.(3)In the third step, the amount of unlabeled data remains stable, and the detection model keeps being trained until the detection accuracy remains stable.

The BA-SSL framework is based on pseudo labels. While being trained on the unlabeled data, the detection model infers and selects pseudo labels that will be back-propagated online to train the model for the unlabeled data by a threshold θ, which is generated through the proposed dynamic threshold strategy. Furthermore, the batch-adding strategy is used in the second stage to avoid the impact on the detection model by a huge amount of unlabeled data.

**Only labeled data.** In the first step, the detection model is trained on the labeled data from the target dataset and the data from the auxiliary dataset. The samples from the target dataset are used as positive samples, which contain several defects in them. The data from the auxiliary dataset are used as negative samples, which do not contain defects. Furthermore, the samples from the auxiliary dataset are augmented through the proposed data augmentation strategy. In this step, the detection model will be trained for about 200 epochs until its detection accuracy remains stable.

**Unlabeled data with an increasing amount.** In this step, the unlabeled data are put into the training process, which can be leveraged through pseudo labels. The pseudo labels are generated online for the unlabeled data by the detection model. While being trained, the detection model predicts bounding boxes and corresponding classification scores for the unlabeled data. The top-scoring bounding boxes whose classification scores are larger than a confidence threshold θ are selected as the pseudo labels that contain bounding boxes and classification scores.

As the classification scores vary depending on the training process, a stable threshold cannot thoroughly and accurately select correct bounding boxes. Therefore, this proposes a dynamic threshold strategy to address this problem. As shown in Equation (1):(1)$$\theta =\underset{\;}{\min}\left(\theta_{0},s_{i}\right)$$
where θ is the threshold used to select pseudo labels, $\theta_{0}$ is a threshold given by the researchers in advance by experience, and $s_{i}$ is the ith largest classification score. The value of i is given according to the statistical result of labeled data of the target dataset. The main idea of this strategy is to use the top-i scoring bounding box’s classification score to replace the original threshold $\theta_{0}$ when the classification scores are so low that all of them are smaller than the threshold $\theta_{0}$ and the model cannot generate enough pseudo labels for the unlabeled data. Once there is not any pseudo label, this unlabeled sample will be seen as a negative sample by mistake, which will disturb the training process. The proposed strategy can efficiently address this problem.

A huge amount of unlabeled data that is directly added into the training process has a tremendous impact on the model and will make the detection model hard to fit. As shown in Figure 5, the detection accuracy dramatically decreases after a huge amount of unlabeled data is added to the training process. Therefore, this work proposed a batch-adding strategy for unlabeled data. The process is shown in Figure 6. The unlabeled data are divided into small batches, and every batch contains the same number of samples. While training the model, the small batches are put into the training process one by one. After a batch is put into training, the amount of unlabeled data that remains is kept stable for a given number of epochs while training the model. This process will recycle until all the unlabeled data are used to train the detection model.

**A stable amount of labeled and unlabeled data.** In this step, the model is trained on a stable amount of labeled and unlabeled data until its detection accuracy remains stable. Though the average accuracy stops increasing, the detection accuracy varies depending on the epochs. Therefore, it needs to keep being trained on the labeled and unlabeled data with the proposed dynamic threshold until the detection accuracy remains stable. Usually, researchers may let the model train for a given number of epochs.

### 3.3. Data-Expanding Strategy

In this work, this data-expanding strategy was proposed to generate negative samples to expand the target dataset by using samples from auxiliary datasets, which contains four steps: (1) clip, (2) filtrate, (3) sharpen, and (4) binarize and smooth. The whole process is shown in Figure 7.

First, the images of auxiliary datasets are clipped into small pitches whose size is the same as that of the images of the target dataset. It should be noticed that these small pitches are defect-free and do not contain defects to avoid disturbance by unexpected types of defects. In this work, the size of images in the target dataset is 640×640, whereas the images in the auxiliary dataset are with a width of more than 3000 pixels. Therefore, this strategy can generate about 3000 alternative images.

Then, the color images are transferred to gray images. After that, a mean blur is used to filter the noise in images. As Equation (2) shows,
(2)$$g\left(i,j\right)=\underset{k,l}{\sum }f\left(i+k,\;j+l\right)h\left(k,l\right)$$
where i and j are the coordinates of the pixel in the processed image, k and l are the coordinates of the kernel, $f\left(i+k,\;j+l\right)$ is the value of the pixels in the original image, $g\left(i,j\right)$ is the value of the pixels in the processed image, and $h\left(k,l\right)$ is the value of the kernel. The mean blur uses a kernel to calculate the mean value of the pixels around the center point and uses the mean value to replace the center point. In this work, the value of the kernel is shown in Equation (3):(3)$$h=\frac{1}{n^{2}}\left[\begin{array}{ccc} 1 & \cdots & 1\\ \vdots & \ddots & \vdots \\ 1 & \cdots & 1 \end{array}\right]\in R^{n\times n}$$
where n is the size of the kernel. In this work, n=7.

Though a mean filter can remove the noise and some other disturbances, it will make the images blurrier and lose some information. Thus, this work used an unsharpen mask (USM) method to sharpen the images. The principle is using a filter to blur the original image and calculates the weighted mean of the original and blurred images. The function is shown in Equation (4):(4)$$s=\left(f-w*G\left(f\right)\right)/\left(1-w\right)$$
where f is the original image, s is the sharpened image, G() is the function of the Gaussian blur, and w is the weight given by researchers. Compared with the mean blur, the Gaussian blur calculates the weighted mean value of the pixels around the center point rather than the mean value. The weight of every pixel follows the Gaussian distribution, as shown in Equation (5):(5)$$H_{i,j}=\frac{1}{2\pi \sigma^{2}}e^{-\frac{{\left(i-k\right)}^{2}+{\left(j-k\right)}^{2}}{2\sigma^{2}}}$$
where $H_{i,j}$ is the template value, with the size of the Gaussian kernel given by $\left(2k-1\right)\times \left(2k-1\right)$. The final kernel needs to be normalized, and the final value is shown in Equation (6):(6)$$h\left(i,j\right)=\frac{1}{\sum_{k,l}H_{k,l}}H_{i,j}$$

Next, the researchers use a threshold to binarize the sharpened images. The threshold is given by experience. At that time, the images are transferred into binary images, which are the same as the images in the target dataset. However, the edge of the items in the images is rough. Therefore, the researchers make use of graphic operations, erosion, and dilation to smooth the edge. Erosion is used to expand the black region, and dilation is used to expand the white region. A group of alternating erosion and dilation can efficiently smooth the binary images.

After being processed, the samples are similar to the samples in the target dataset and can be used as negative samples to expand the target dataset.

## 4. Experiments

In this section, experiments are performed to analyze the proposed DE-SSD and evaluate its efficacy by comparing the proposed method with other PCB defect detection models. To evaluate the proposed method’s efficacy, this paper tested the proposed DE-SSD on the target dataset DeepPCB with different numbers of labeled samples. The experiment has two parts: a semi-supervised part and a fully supervised part. A semi-supervised part is used to show the performance of the proposed method with a few labeled samples and plenty of unlabeled samples, and a fully supervised part is used to show the performance of the proposed method with abundant labeled data.

### 4.1. Implementation Details

In this work, the researchers used YOLOv5x to verify the proposed method, which is also the baseline model, which has 87.7 M parameters and 284 layers. The researchers deployed the proposed method on a server, which contains two Nvidia RTX2080Ti GPUs. The operating system is Linux 3.10.0-862.el7.x86_64. The main programming language is Python 3.7.11, with the installation of PyTorch 1.10.2, Torchvision 0.11.3, and other Python libraries such as NumPy 1.21.5.

As aforementioned, the target dataset DeepPCB contains 1500 labeled samples, in which 1000 samples are used as a train set and 500 samples are used as a test set. The auxiliary dataset PKU-PCB contributes 3000 negative samples, and the researchers selected 1500 negative samples to expand the target dataset. For the semi-supervised part, the researchers randomly selected 50, 100, and 200 as the labeled data, and the others are used as the unlabeled data. The unlabeled data were divided into batches, each of which contained the same number of samples as the labeled samples. For the fully supervised part, 1000 labeled samples in a train set with 1500 negative samples were all used to train the detection models. By the way, the number of negative samples was always kept at 1.5 times as many as the total number of defect samples, including the labeled and unlabeled samples.

The set of hyperparameters is following the literature and the source code. The researchers trained the detection model using an SGD optimizer with an initial learning rate of $10^{-3}$, 0.0005 weight decay, and a batch size of 16. For the semi-supervised experiment, the first and third training stages lasted for 400 epochs. In the second stage, the number of epochs when the amount of unlabeled data was kept stable was 120. The researchers used mAP for evaluation, and the set of mAP is following the literature [14].

To compare the proposed DE-SSD with other methods, the researchers selected several classical PCB defect detection models. In the fully supervised part, this paper used I.P. [29], RDDN [15], and several classical object detection models, such as YOLOv1 [20] and Faster-RCNN [30]. In the semi-supervised part, this paper used SSD [14] and YOLOv5, of which YOLOv5 is the baseline model. These models were trained on manners proposed in the literature. To evaluate the methods, the researchers used three metrics: mAP, precision, and recall. The precision is the proportion of the true defects in the detected defects, and the recall is the proportion of the true defects that are detected. These two metrics reflect the cost of the misclassification and the proportion of missed defects. The mAP is the average of precisions with different recalls, which can reflect the detection accuracy of the detection models. As the main evaluation metric of the dataset is mAP [14], most of the previous works only gave the mAP results. Thus, the researchers used mAP to compare the proposed method with other methods and used precision and recall to compare the proposed method and the baseline model.

### 4.2. Results

The experiment results are shown in Table 1 and Table 2. In addition, some detection results are shown in Figure 8. In Figure 8, the right part shows the original samples, and the left part shows the detection instances. The detection results show that the proposed method can accurately and fully detect defects in samples.

**Comparison with the baseline model.** To furtherly show the efficiency of the proposed method, this paper compared the performance of the DE-SSD method with that of the baseline model. As shown in Table 2, the precisions and recalls of the DE-SSD are always higher than those of the baseline model. Especially on 50-label samples, the recall of DE-SSD is 7.8% higher than that of YOLOv5, and the precision is 5.1% higher than that of YOLOv5. The improvement of recall can efficiently reduce the cost of missed detection, and the improvement of precision can significantly reduce the cost of misclassification. Compared with the baseline model, DE-SSD can reduce both the false-positive and false-negative rates. These results verify the proposed method’s effectiveness.

**Semi-supervised part results.** This part compares the performance of the proposed DE-SSD method with other PCB defect detection models. The detection accuracies of DE-SSD on all numbers of labeled data are higher than those of other methods, especially when the number of labeled samples is small. The proposed DE-SSD achieves mAP 98.4% on 50 labeled data, which is higher than that of other methods on 1000 labeled samples, except SSD.

Compared with the baseline model YOLOv5, DE-SSD shows a great improvement, which is larger than the improvement between SSD_full_ and SSD_semi_, with about mAP 1.9–4.7% and mAP 1.4–1.8%, respectively. As a result, the detection accuracies of DE-SSD on 50, 100, and 200 labeled samples focus on mAP 98.4–98.7%, which are much higher than those of SSD and close to the detection accuracy of DE-SSD on 1000 samples.

**Fully supervised part results.** This part compares the performance of the proposed DE-SSD method with other defect detection methods. As shown in Table 1, the detection accuracy of the method proposed in this work is the highest.

Compared with the classical object detection model such as YOLOv1 and Faster-RCNN, the proposed method achieves significant improvements, with mAP 6.1% and 1.1%, respectively. Compared with the defect detection models such as I.P. and RDDN, the detection accuracy of the proposed method is much higher, with mAP 9.4% and 1.8% improvements, respectively. Compared with SSD, this work only achieved an improvement with mAP 0.1%. However, SSD needs both defect samples and the corresponding template samples to detect defects in PCB images, which has a high requirement for the training and testing data and limits its usage and generality. In contrast, the proposed DE-SSD only needs the defect samples to detect defects.

**Summary.** Compared with other methods, the proposed DE-SSD has three advantages: (1) it has a high detection accuracy and is the SOTA method; (2) it can only use a few labeled samples to train the detection model, which reduces the need for labeled samples; and (3) it achieves high detection accuracy on a small number of labeled samples, which is similar to that on a large number of labeled samples.

## 5. Ablation Study

This section performs an ablation study on the key components of DE-SSD. The study performs experiments to analyze the impact on the performance of the detection model by the influence of (1) the BA-SSL framework, (2) DE strategy, and (3) batch-adding strategy for unlabeled data. In addition, the researchers analyze the necessity of the dynamic threshold strategy in a few words.

### 5.1. Batch-Adding Strategy

This section evaluates the influence of the batch-adding strategy by comparing the detection accuracies of the proposed DE-SSD and DE-SSD without the batch-adding strategy.

As shown in Table 3, the detection accuracies of the DE-SSD model are higher than that of DE-SSD without the batch-adding strategy. For example, DE-SSD achieves mAP 98.4%, 98.6%, and 98.7% on 50, 100, and 200 samples, respectively, whereas DE-SSD without batch-adding strategy achieves 97.8%, 98.1%, and 98.5%, respectively, which is a significant improvement. This method is especially efficient when the number of labeled defect samples is small because, at that time, the unlabeled samples more severely disturb the training process.

### 5.2. BA-SSL Framework

This section evaluates the influence of the BA-SSL framework by comparing the detection accuracies of the original YOLOv5, YOLOv5 with the BA-SSL framework, YOLOv5 with the proposed DE strategy, and the proposed DE-SSD. The researchers train these models and evaluate them. The experiments are performed on the aforementioned datasets, too. Notably, YOLOv5 with the BA-SSL framework is trained with a limited number of unlabeled samples, whereas DE-SSD is trained with all the unlabeled samples.

As shown in Table 4, the detection accuracies of the proposed DE-SSD are higher than those of YOLOv5 with the DE strategy. The detection accuracies of DE-SSD are mAP 98.4%, 98.6%, and 98.7%, whereas YOLOv5 with DE strategy achieves mAP 90.5%, 93.5%, and 96.2%, which is a significant improvement. Moreover, the detection accuracy of the YOLOv5 model will improve if trained with unlabeled data and the BA-SSL framework. These experiments show the efficiency and necessity of the BA-SSL framework.

However, the ability of the BA-SSL framework to leverage the unlabeled samples is still limited. If there are too many unlabeled samples, the training process will be disturbed a lot. As shown in Figure 9, YOLOv5 with the BA-SSL framework cannot fit when trained with all the unlabeled samples. Thus, it requires the researchers to manually limit the number of unlabeled samples.

### 5.3. Data-Expanding Strategy

This section evaluates the influence of the proposed DE strategy by comparing the detection accuracies of YOLOv5 models trained in different manners.

As shown in Table 4, the detection accuracies of the proposed DE-SSD are much higher than those of DE-SSD without DE strategy, i.e., YOLOv5 with the BA-SSL framework. DE-SSD achieves mAP 98.4%, 98.6%, and 98.7% on 50, 100, and 200 samples, respectively, whereas YOLOv5 with the BA-SSL framework achieves mAP 95.8%, 96.6%, and 97.3%, respectively. It is a significant improvement because the DE strategy can make the model able to leverage a larger amount of unlabeled data. The negative samples can limit the proportion of, and avoid the impact of, unlabeled samples. Thus, DE-SSD can train the detection model well with all the unlabeled data, which improves the model’s performance.

However, as shown in Table 4, the DE strategy is not always effective without the BA-SSL framework. The detection accuracy of YOLOv5 dramatically decreases if trained on small numbers of labeled samples with the DE strategy. Because even though the data augmentation strategy reduces the difference between negative and defect samples in backgrounds, it can still influence the training process especially when there are not enough defect samples.

### 5.4. Discussion

These experiments evaluate the influence of different components of the proposed method, including the BA-SSL framework, DE strategy, and batch-adding strategy. These components all contribute to the final result. Thus, they are necessary. However, the BA-SSL framework and DE strategy have their limits when used alone.

The BA-SSL framework is always effective in any training manner as it can always improve the model’s performance on the target dataset. However, it cannot leverage too many unlabeled samples when there are only a few labeled defect samples, which will disturb the training process and cause underfitting. On the other hand, the DE strategy can improve the model’s performance on all numbers of labeled samples with unlabeled samples. However, if there are only a few defect samples without unlabeled samples, this DE strategy may disturb the training process and decrease the detection accuracy. The DE strategy was originally proposed to improve the detection accuracy on a large number of labeled samples, and the BA-SSL framework was proposed to improve the detection accuracy on a small number of labeled samples.

Though these two methods have their limits, the experimental results show that they can work together well and improve detection accuracy a lot, especially on small numbers of labeled samples. This is because they can eliminate the disturbance of each other. When the number of labeled samples is small, the BA-SSL framework can leverage the huge amount of unlabeled data to improve the number of defect samples, which can avoid the disturbance by the DE strategy. In the meantime, the DE strategy can generate a large number of negative samples and limit the proportion of unlabeled data, which will improve the model’s ability to leverage unlabeled samples and avoid disturbance by unlabeled data. As a result, the detection accuracies on small numbers of labeled samples are improved a lot, and the gap between accuracies on small numbers of labeled samples and large numbers of labeled samples is much smaller.

## 6. Conclusions and Future Work

Though the PCB defect detection methods have made significant strides, semi-supervised learning is more suitable for PCB defect detection, which uses both labeled and unlabeled samples and saves the cost of data labeling. However, the detection accuracy cannot meet the needs of practical applications. Thus, this paper developed a semi-supervised defect detection method with a data-expanding strategy, containing an object detection model, BA-SSL framework to leverage the unlabeled data, and DE strategy to integrate the target and auxiliary datasets. The result of the experiments shows the efficacy of the proposed DE-SSD method as it is the SOTA method on the DeepPCB dataset. Compared with the baseline model, the proposed method improves precision and recall especially when the number of labeled samples is small, which reduces the cost of misclassification and the number of defects that are undetected. The performance of the proposed method will provide an idea for improving the detection accuracy of PCB defect detection models by applying unlabeled data and leveraging the auxiliary datasets.

Though the proposed DE-SSD method demonstrates impressive performance, there are still several limitations. First, DE-SSD needs researchers to develop a DE strategy according to the characters of the auxiliary dataset. Second, the training process is disturbed by label noise in the pseudo labels, which limits the detection accuracy. Third, the selected PCB datasets only contain seven types of defects but do not consider many other types of defects, such as dotted defects, line defects, and so on. Therefore, for future work, researchers should find a general method for DE and a label-denoising method for semi-supervised PCB defect detection. Furthermore, the researchers should collect more kinds of samples and develop the method furtherly aiming at new kinds of defects.

## Figures and Tables

**Figure 1 sensors-22-07971-f001:**
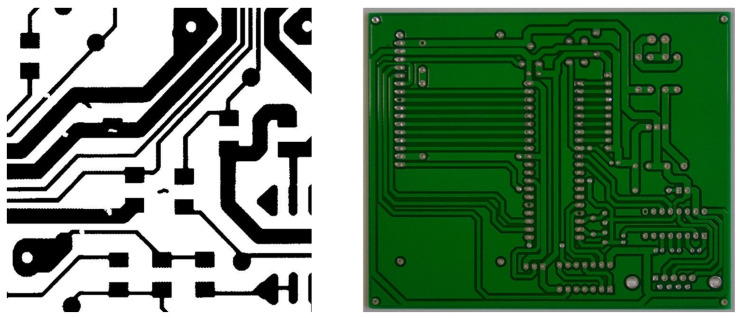
Two images from different datasets. Left is from DeepPCB and right is from PKU-PCB.

**Figure 2 sensors-22-07971-f002:**
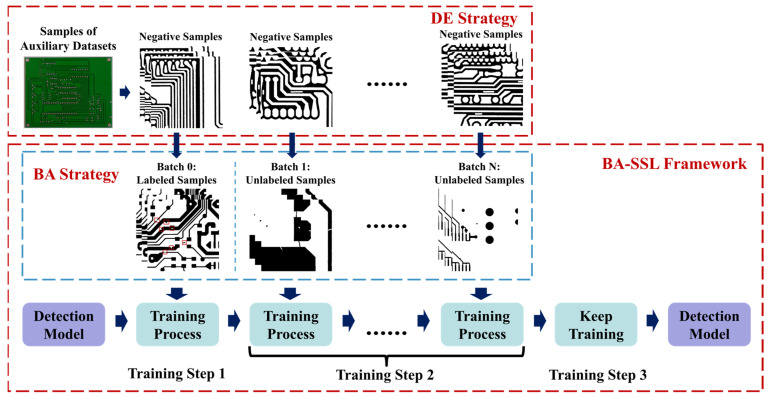
Training process of proposed DE-SSD method.

**Figure 3 sensors-22-07971-f003:**
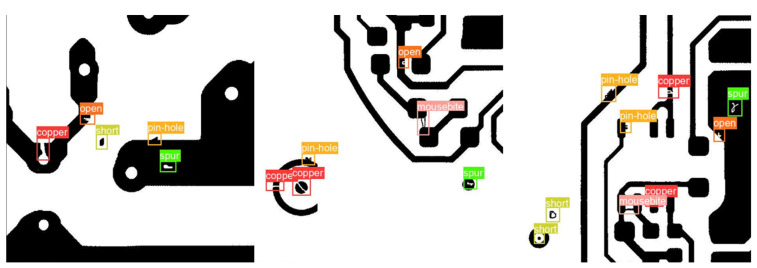
Defect samples that contain bounding boxes and classes of detects.

**Figure 4 sensors-22-07971-f004:**
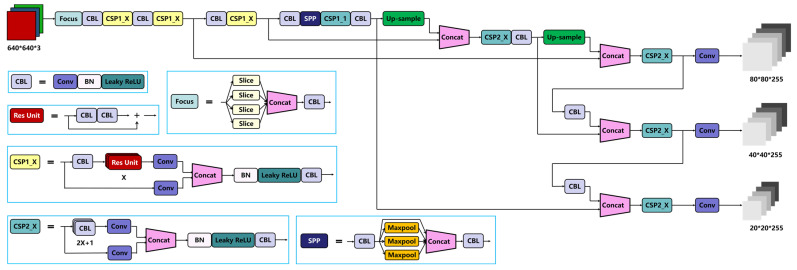
Structure of YOLOv5.

**Figure 5 sensors-22-07971-f005:**
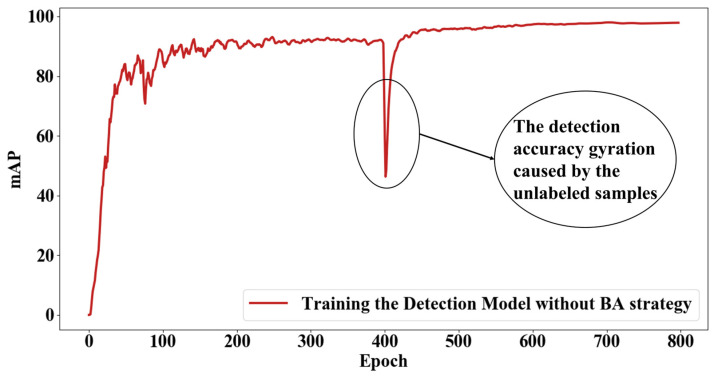
mAPs of each epoch in training process without batch-adding strategy.

**Figure 6 sensors-22-07971-f006:**
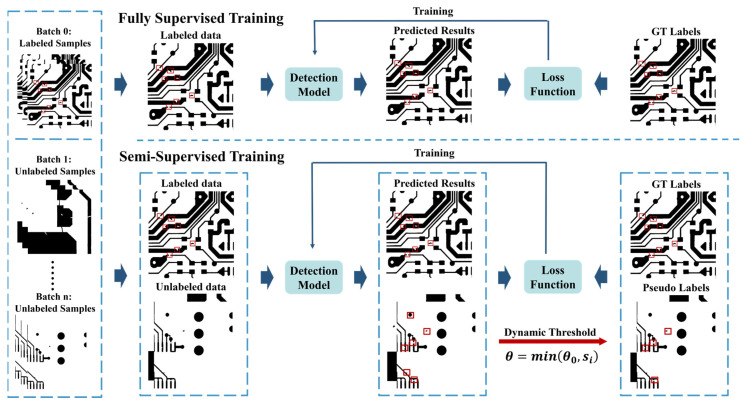
Training process of proposed BA-SSL framework.

**Figure 7 sensors-22-07971-f007:**
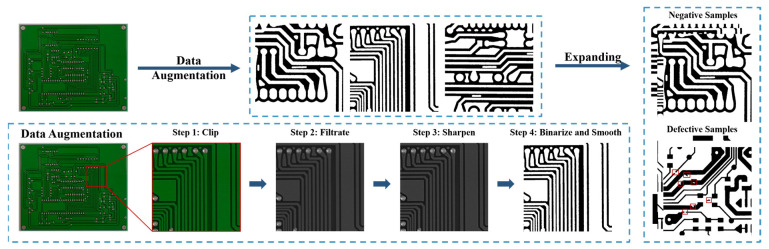
Whole process of proposed data-expanding strategy.

**Figure 8 sensors-22-07971-f008:**
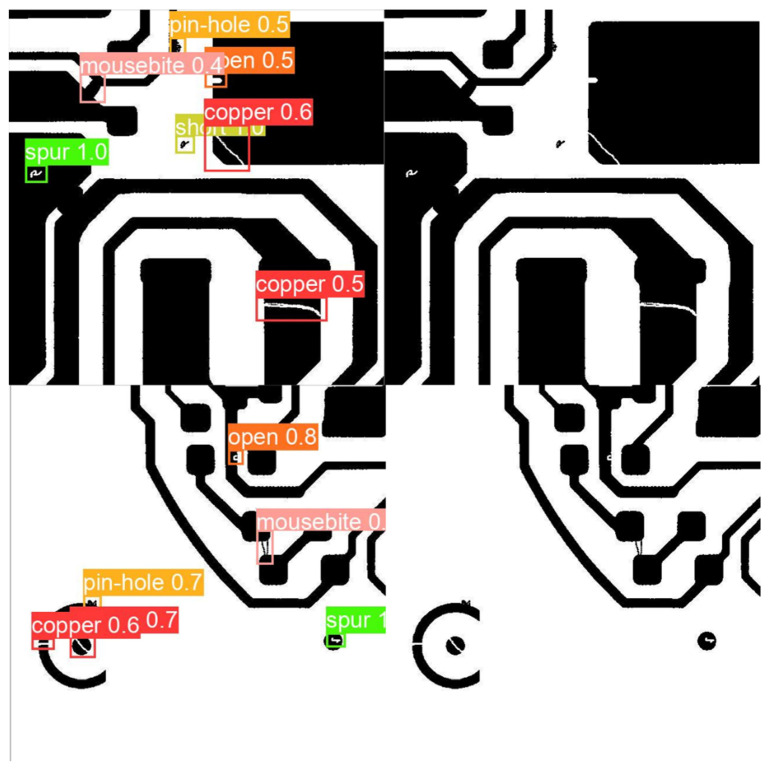
Detection results of proposed DE-SSD method.

**Figure 9 sensors-22-07971-f009:**
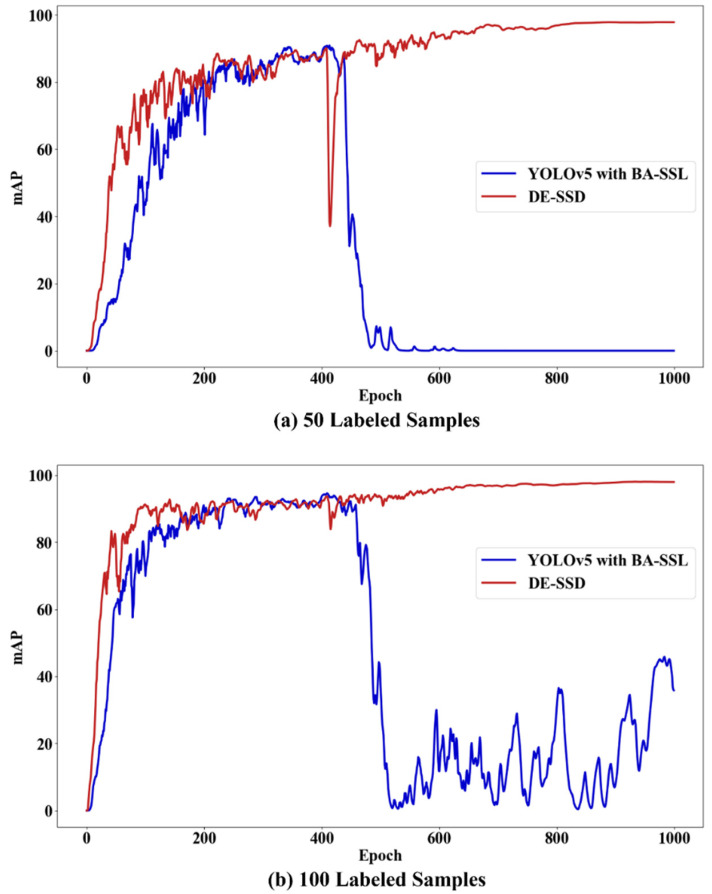
mAP of each epoch while training the detection model. (**a**) Trained with 50 labeled samples. (**b**) Trained with 100 labeled samples. These models are trained with all unlabeled samples.

**Table 1 sensors-22-07971-t001:** Detailed information on experiment results.

Fully Supervised Part				
Method	50 Samples	100 Samples	200 Samples	1000 Samples
I.P. [29]	-	-	-	89.3%
YOLOv1 [20]	-	-	-	92.6%
RDDN [15]	-	-	-	96.9%
Faster-RCNN [30]	-	-	-	97.6%
**Semi-Supervised Part**				
**Method**	**50 Samples**	**100 Samples**	**200 Samples**	**1000 Samples**
SSD_full_ [14]	86.5%	90.7%	92.1%	98.6%
SSD_semi_ [14]	89.3%	92.4%	93.5%	98.6%
YOLOv5	93.7%	95.4%	96.8%	98.4%
**YOLOv5 + DE-SSD**	**98.4%**	**98.6%**	**98.7%**	**98.7%**

**Table 2 sensors-22-07971-t002:** Detailed information on comparison between DE-SSD method and baseline model.

Number of Samples	Method	Precision	Recall	mAP
**50**	YOLOv5	92.1%	88.8%	93.7%
	**YOLOv5 + DE-SSD**	**97.2%**	**96.6%**	**98.4%**
**100**	YOLOv5	94.3%	93.1%	95.4%
	**YOLOv5 + DE-SSD**	**97.8%**	**96.4%**	**98.6%**
**200**	YOLOv5	96.2%	94.8%	96.8%
	**YOLOv5 + DE-SSD**	**98.1%**	**96.8%**	**98.7%**
**1000**	YOLOv5	97.9%	95.5%	98.4%
	**YOLOv5 + DE-SSD**	**98.1%**	**96.8%**	**98.7%**

**Table 3 sensors-22-07971-t003:** Testing results of DE-SSD without batch-adding strategy and DE-SSD on different amounts of labeled samples.

Method	50 Samples	100 Samples	200 Samples
DE-SSD without batch-adding	97.8%	98.1%	98.5%
**DE-SSD**	**98.4%**	**98.6%**	**98.7%**

**Table 4 sensors-22-07971-t004:** Testing results of original YOLOv5, YOLOv5 with proposed BA-SSL framework, YOLOv5 with proposed DE strategy, and proposed DE-SSD on different amounts of labeled samples.

Method	50 Samples	100 Samples	200 Samples
YOLOv5	93.7%	95.4%	96.8%
YOLOv5 + DE	90.5%	93.5%	96.2%
YOLOv5 + BA-SSL	95.8%	96.6%	97.3%
**YOLOv5 + DE-SSD**	**98.4%**	**98.6%**	**98.7%**

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
