# Peer review of "Semi-Supervised Defect Detection Method with Data-Expanding Strategy for PCB Quality Inspection"

_sensors, 2022, doi:10.3390/s22207971_

Round 1

Reviewer 1 Report

The research of this paper has certain practical significance and application value for PCB defect detection. Using the method proposed in the article, people can only use a few labeled samples to train the detection model without loss accuracy, which reduces the need and cost for labeled samples. In general, the format of the article is standardized, clear, and the writing is smooth and the logic is relatively strict.

Suggestions:

1. The format of the literature citation is incorrect. The square brackets should be added to the corresponding numbers.

2. The serial numbers of the five parts in the 132-line paragraph are all "0". Also, are the names of the two lines in Figure 7 wrong? The word "EE-SSL" does not appear throughout the article.

3. There are some spelling and punctuation errors, line 290, the comma in "dilation, to" in line 336 should be removed, and "Ablution" in line 411 should be "Ablation".

4. The dataset only contain open, short, mouse bite, spur, spurious copper, and pinhole 7 types of defects. However, and many other types of defects have not been considered, such as dotted defects, line defects, block defects, and so on.

5. The samples in the benchmark dataset has not been introduced for each types of defects. And the final result has not given defects detection instance. 

Reviewer 2 Report

Researchers propose a semi supervised defect detection method for printed circuit boards. The proposed approach is creative it has potential to improve the accuracy. However, the increase in the accuracy does not appear to be a significant amount. The accuracy is not meaningful alone, while cost of false postiive and false negative could be different. While an algortihm produces a better accuracy, its cost of misclassfication may be more than a less accurate model. I suggest authors to use other measures as well to check the cost of misclassification.

There are also some minor issues listed below: 

-Figure 2 should be placed on somewhere close to its first mention.

-What is modern manufacturing 112?

-There is a problem about citing numbers.

Round 2

Reviewer 2 Report

I see that the researchers has considered my concerns and it is acceptable.